# CAN LARGE LANGUAGE MODELS EXPRESS UNCERTAINTY LIKE HUMAN?

## ABSTRACT

Large language models (LLMs) are increasingly used in high-stakes settings, where overconfident responses can mislead users. Reliable confidence estimation has been shown to enhance trust and task accuracy. Yet existing methods face practical barriers: logits are often hidden, multi-sampling is computationally expensive, and verbalized numerical uncertainty (e.g., giving a 0–100 score) deviates from natural communication. We revisit linguistic confidence (LC), where models express uncertainty through hedging language (e.g., probably, might), offering a lightweight and human-centered alternative. To advance this direction, we 1) release the first diverse, large-scale dataset of hedging expressions with human-annotated confidence scores, and 2) propose a lightweight mapper that converts hedges into confidence scores at near-zero cost. Building on these resources, we 3) conduct the first systematic study of LC across modern LLMs and QA benchmarks, revealing that while most LLMs underperform in expressing reliable LC, carefully designed prompting achieves competitive calibration and discriminability. Finally, we 4) introduce a fine-tuning framework that further improves LC reliability. Taken together, our work positions linguistic confidence as a scalable, efficient, and human-aligned approach to LLM uncertainty estimation, and calls for deeper exploration of this promising yet underexplored direction. The code and dataset are anonymously available at `https://anonymous.4open.science/r/Linguistic-Uncertainty-Dataset-051E`

## 1 INTRODUCTION

Large language models (LLMs) are increasingly deployed in real-world applications, from education and healthcare to law and scientific discovery. While their capabilities make them powerful assistants, LLMs are also prone to hallucinations and factual errors, and human overreliance on their outputs can lead to serious consequences. For instance, a U.S. lawyer once submitted fabricated cases generated by ChatGPT, resulting in professional sanctions (ABC News, 2023). Recent social experiments demonstrate that people adjust their reliance on AI depending on how confident the model appears: reliable expressions of uncertainty can enhance trust, satisfaction, and task accuracy (Kim et al., 2024; Xu et al., 2025). These findings highlight the importance of associating reliable uncertainty estimates with LLM responses to support human decision-making. Ultimately, the conveyance of confidence plays a central role in shaping trust and guiding human–AI interaction.

A growing body of work explores the extraction and representation of confidence in LLM outputs. One line of research leverages white-box token probabilities, such as token-level likelihoods or perplexity (Malinin & Gales, 2020). These methods are simple and inexpensive but require access to model logits, which are typically unavailable in commercial LLM APIs. The second line of work estimates confidence through multiple generations, using techniques such as semantic entropy (Kossen et al., 2024a; Farquhar et al., 2024), self-consistency (Wang et al., 2022), bayesian semantic confidence (Lamb et al., 2025) or p(true) scoring (Kadavath et al., 2022). While often effective, these approaches are computationally expensive, as they require multiple model calls or auxiliary networks, limiting their practicality. The third approach directly elicits numerical probabilities from the model, prompting it to output a confidence score alongside its answer (Tian et al., 2023; Xiong et al., 2023; Lin et al., 2022). However, such scores rarely align with common user behavior or natural communication, as users do not typically phrase queries with explicit instructions like "Please output your confidence along with the answer." Recent work has therefore argued for more human-

centered forms of uncertainty communication (Devic et al., 2025). Taken together, these limitations motivate the search for uncertainty representations that are both efficient and user-friendly.

Linguistic confidence—expressing uncertainty through natural language (e.g., hedges such as probably, might, or I am not entirely sure)—offers a promising alternative. It integrates seamlessly into responses, requires minimal computational overhead, and mirrors how humans naturally convey uncertainty. Nevertheless, research on linguistic confidence remains underdeveloped. The most relevant study is by Yona et al. (2024), which examined the alignment between LC and semantic consistency among multiple generations. However, this work does not directly evaluate LC under standard uncertainty estimation criteria such as calibration and discriminability. Moreover, their approach maps answers to numerical confidence scores using an LLM-based process that was only validated on the small dataset, which contains merely 18 uncertainty phrases (Flanagan & Fagen-Ulmschneider, 2019), thereby limiting the reliability of their findings. In addition, LLM-based judging of confidence phrases is prohibitively expensive, costing around $3 per evaluation round when applied to modern benchmarks like SimpleQA (Wei et al., 2024).

In this work, we revisit linguistic confidence as a practical and scalable approach to LLM uncertainty estimation. Our contributions are fourfold. **First**, we construct a large human-annotated dataset of linguistic uncertainty expressions, capturing diverse hedging beyond fixed phrase lists and enabling systematic evaluation of how well confidence mapping aligns with human judgments. **Second**, we introduce a lightweight mapper that efficiently converts hedging expressions into confidence scores at near-zero cost. Unlike the costly and time-consuming LLM judging through API calls adopted in (Yona et al., 2024), our mapper incurs negligible latency, making it suitable for applications with strict real-time requirements. **Third**, we conduct a comprehensive evaluation of LC across state-of-the-art LLMs on benchmarks of varying difficulty, including SimpleQA (Wei et al., 2024), PopQA (Mallen et al., 2022), and NQ-Open (Lee et al., 2019). This analysis covers both calibration and discriminability, showing that most SOTA LLMs, such as GPT-5, perform poorly at linguistic confidence; yet with carefully designed prompts, LC can achieve performance comparable to other methods (Farquhar et al., 2024; Xiong et al., 2023). **Finally**, we propose a fine-tuning framework that enhances LC through supervised fine-tuning. Specifically, we fine-tune a Qwen-8B model on our constructed dataset and evaluate it across multiple QA benchmarks, where results consistently demonstrate improvements in both calibration and discriminability.

## 2 RELATED WORK

**Confidence Estimation in LLMs** Confidence estimation in LLMs has been studied from multiple perspectives. A widely-used approach leverages token probabilities, with perplexity (Malinin & Gales, 2020) being a representative method. Kadavath et al. (2022) further propose P(True) using self-evaluation and the probability of an affirmative response, often by inspecting token probabilities of the "true" token. However, these confidence estimators are subject to token probability availability, making them inapplicable to many closed-source LLMs. To overcome such limitations, a series of black-box methods are proposed. Verbalized confidence is one of the early endeavors, which prompts models to articulate their confidence explicitly in numerical or ordinal forms, offering efficiency and generality (Xiong et al., 2023; Lin et al., 2022), though it lacks human conversational intuition. Another class of black-box methods involves repeated sampling, such as self-consistency (Wang et al., 2022) measuring the agreement among thinking paths. Similarly, Farquhar et al. (2024) present semantic entropy that measures the semantic consistency of answer clusters from multiple generations. These semantics-focused estimations also inspired hybrid approaches incorporating perplexity with semantic clusters (Lamb et al., 2025) and the Monte Carlo approximation of P(True) (Farquhar et al., 2024). Nevertheless, these methods remain constrained by substantial computational costs. As such, semantic entropy probes (Kossen et al., 2024b) address the significant overhead by learning a semantic entropy estimator from hidden activations, though their applicability is limited to models with accessible hidden states.

**Linguistic Confidence Estimation** Human communication often conveys uncertainty through hedging, yet existing methods remain poorly aligned with natural expressions. Mielke et al. (2022) pioneered the use of LC in training by categorizing responses as DK (don't know), LO (uncertain), or HI (confident). While influential, this coarse taxonomy oversimplifies the nuanced nature of hedging and predates modern LLMs. More recently, Yona et al. (2024) provided the first sys-

tematic study of LLMs' ability to hedge, finding that LC fails to align with semantic consistency regardless of prompting. In contrast, our experiments show that carefully designed prompts can yield strong calibration and discriminability. Moreover, their reliance on a fixed list of 18 hedging terms (Flanagan & Fagen-Ulmschneider, 2019) limits generalizability. Likewise, Wang et al. (2024) extend existing measures of uncertainty and propose a post-hoc calibration method, but again only for a fixed set of hedging terms. Complementary work by Belem et al. (2024) further demonstrates that mapping hedging expressions to numeric scores is highly sensitive to LLM priors, undermining reliability. These limitations underscore that LC in LLMs remains underexplored, calling for more robust mapping methods and comprehensive evaluation.

## 3 BUILD CONFIDENCE DATASET

To construct a human-annotated dataset, we first **(1)** collect uncertain expressions generated by LLMs, **(2)** obtain human annotations through the Amazon Mechanical Turk platform and filter responses using validation questions and **(3)** identify reliable annotators who consistently follow instructions. **(4)** Subsequently, we determine the upper and lower confidence bounds for valid responses and apply these screening criteria to all annotators' responses. **(5)** Finally, we retain only those expressions with at least three valid annotations for inclusion in the test benchmarks. The full benchmark building process is presented in Figure 1.

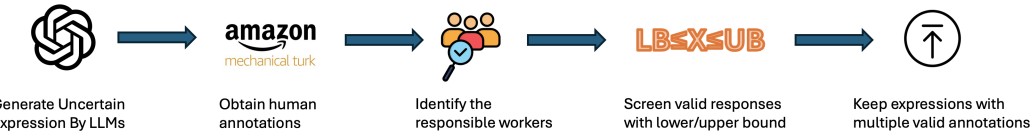

Figure 1: An illustration of the benchmark building process.

**Step 1: Collecting Uncertain Expressions.** First, we generate uncertain responses for 200 questions uniformly sampled from SimpleQA (Wei et al., 2024) by prompting four LLMs: GPT-5, Grok-3, Claude-Sonnet-4, and Gemini 2.5 Pro. An example prompt is provided in Appendix B.1. Specifically, the LLMs are instructed to generate 10 expressions for each of the five different confidence levels, ranging from high confidence, moderate confidence, low confidence, lowest confidence, to complete uncertainty. This process yields $4 * 10 * 5 * 200 = 40,000$ uncertainty expressions, from which we uniformly sample 10,000 for building the benchmark.

**Step 2: Annotations from Amazon Mechanical Turk.** To obtain annotations, we design a survey consisting of 105 expressions, where 100 are real tasks and 5 are validation items pre-annotated by human experts. Annotators are asked to provide a confidence score between 0 and 100 for each expression. If their scores deviate substantially from the expert annotations[1].(e.g., assigning a score of 99 to "I guess the answer is ..."), all 105 of their responses are rejected. An example survey is shown in Figure 6. Each uncertainty expression is annotated by five different participants, and to promote annotator diversity, each worker is restricted to at most five surveys. Through this process, we collect a total of 50,000 valid annotations, corresponding to five confidence scores per expression.

**Step 3: Post-Survey Screening for Response Reliability.** Even though some workers pass the validation questions, they still fail to follow the instructions carefully, potentially leading to a noisy benchmark. For example, as shown in the instructions of Figure 6, workers are required to assign a score of 0 to expressions that explicitly reject answering the question, such as "I am sorry, but I can't definitively answer that." However, some workers incorrectly assign a score of 100, which is the opposite of the intended instruction. To address this, we identify responsible workers who follow instructions and use their responses as the basis for further screening. A worker is considered responsible if they consistently assign a score of 0 to such refusal expressions. The distribution of responses from these responsible workers are illustrated in Figure 2.

---

[1]We trained 20 Ph.D. students with clear instructions, each annotating five validation uncertain expressions. A substantial deviation was defined as at least three out of five annotations falling outside two standard deviations from the expert mean. This relatively lenient validation criterion ensured fairer treatment of Amazon Mechanical Turk workers, resulting in an acceptance rate of 90%.

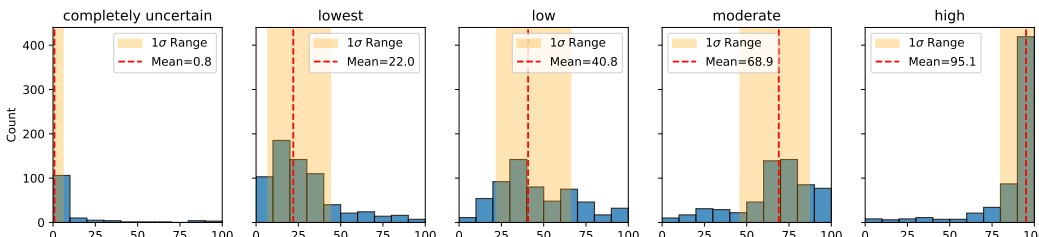

Figure 2: Distribution of annotations from responsible workers. The red dotted line represents the mean, and the shaded area represents a filtering range of $1\sigma$ around the mean.

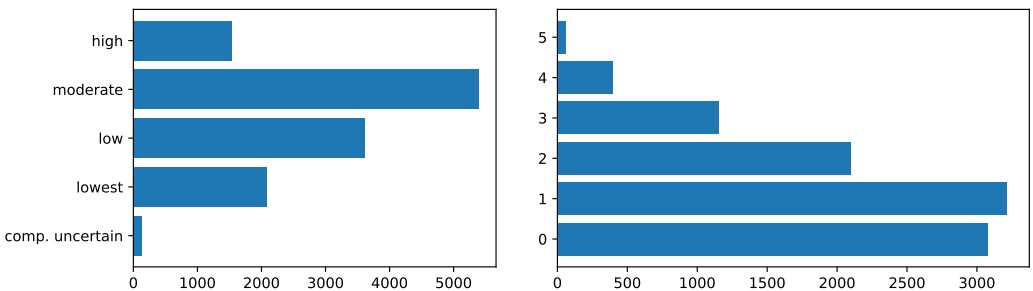

Figure 3: **(a)** Valid annotation distribution **(b)** Expressions with different number of valid annotations

**Step 4: Determining Upper and Lower Bounds for Valid Responses.** Since annotators may occasionally make mistakes due to fatigue, we adopt a neighborhood-based screening strategy. For each confidence level, we first identify the range of 1 standard deviation ($1\sigma$) around the mean. This range defines the upper and lower bounds for acceptable annotations. We then apply these bounds to all responses, retaining those within the valid range. This process results in 12,762 valid annotations, with their distribution across confidence levels shown in Figure 3 (left). Each expression initially has five annotations, but many fall below three valid annotations after filtering. Only a small number of expressions retain more than three valid ones, as illustrated in Figure 3 (right).

**Step 5: Retaining Expressions with Sufficient Valid Annotations.** We retain all expressions with at least three valid annotations as the final test benchmark and take the average score as their final confidence annotation. This yields a dataset consisting of 988 expressions at the moderate level, 428 at the low level, 131 at the lowest level, 64 at the high level, and 11 at the completely uncertain level. We provide some example of the dataset in Table 5 in the Appendix.

## 4 TRAIN A CONFIDENCE MAPPER

To quantitatively evaluate the performance of linguistic confidence, the first step is to reliably map the confidence expressed through hedging language to numerical confidence scores. Prior studies (Yona et al., 2024; Ji et al., 2025) leveraged LLMs to estimate the decisiveness of sentences using carefully designed prompts (see Appendix D.6). The resulting mapping was validated against human probability judgments collected by Flanagan & Fagen-Ulmschneider (2019), which provides 18 probability expressions such as "Almost Certain," "Highly Likely," "Very Good Chance," "We Believe," "Probably," and "About Even." Although this dataset includes responses from over 123 participants, its coverage of uncertainty expressions remains insufficient to represent the diversity observed in natural conversations. For instance, Table 6 illustrates additional hedging strategies identified from an academic writing perspective (Academic Marker, 2025). The restricted diversity of the dataset in Flanagan & Fagen-Ulmschneider (2019) raises concerns about the reliability of the mapping. Furthermore, the high computational cost and latency of LLM-based approaches present substantial barriers for cost-sensitive and real-time applications.

To overcome these limitations, we develop a lightweight model that efficiently and accurately maps sentences to confidence scores. Our mapper uses DistilRoBERTa (Sanh et al., 2019) as the encoder, followed by a linear layer with sigmoid activation. Training is performed under two settings: (1) on

a dataset of LLM-generated sentences obtained via prompt B.2, with labels assigned as {completely uncertain: 0, lowest: 0.25, low: 0.5, moderate: 0.75, high: 1}; and (2) on human-annotated expressions (previously discarded in Step 5 of Section 3 due to fewer than three valid annotations).

For comparison with prior work, we also report the decisiveness scores used by Yona et al. (2024) and Ji et al. (2025) and employ the direct prompt method described in Appendix D.4. Evaluation is conducted on the benchmark introduced in Section 3, using mean squared error (MSE) between predicted scores and average human annotations as the evaluation metric. To ensure fairness, all training data are disjoint from the test set. Results are presented in Table 1.

| Conf. Mapping Method | Base Model | MSE | *Cost (USD) | *Latency (s) |
|---|---|---|---|---|
| Decisiveness Score (D.6) | gpt-5-mini | 385.40 | 0.67 | 551 |
| (Yona et al., 2024) | gpt-5 | 278.39 | 3.12 | 638 |
| | gpt-4o-mini | 351.64 | 0.05 | 654 |
| Direct Prompt (D.4) | gpt-5-nano | 300.36 | 0.02 | 663 |
| | gpt-5-mini | 191.92 | 0.11 | 578 |
| | gpt-5 | 183.23 | 0.53 | 678 |
| Ours | LLM anno. trained | 85.19 | near 0 | 1.32 |
| | Human anno. trained | 50.68 | near 0 | 1.32 |

Table 1: Comparison between different confidence mapping methods. *Cost and Latency are estimated based on the evaluation on SimpleQA dataset with inference API provided by OpenAI.

Our mapper not only achieves substantially lower error than LLM-based baselines but also operates at near-zero cost and low latency on a local machine. This efficiency makes it well suited for integration into cost-sensitive and latency-critical applications, including online training and real-time decision-making systems.

## 5 STUDY ON LINGUISTIC CONFIDENCE

### 5.1 EXPERIMENTAL SETUP

With a reliable confidence mapper in place, we are now able to conduct a systematic analysis of the linguistic confidence performance of current LLMs. To ensure the generality of our findings, we evaluate across multiple QA benchmarks of varying difficulty, including SimpleQA (Wei et al., 2024), NQ-Open (Lee et al., 2019), and PopQA[2] (Mallen et al., 2022). These datasets span both simple factoid queries and more complex open-domain settings, providing a diverse testbed for uncertainty evaluation. For example, the accuracy of gpt-5-mini ranges from about 20% on SimpleQA to nearly 60% on NQ-Open, underscoring the substantial variation in dataset difficulty.

We focus on two complementary dimensions of uncertainty estimation: calibration (Guo et al., 2017) and discriminability (Hendrycks & Gimpel, 2016). Calibration measures how well predicted confidence aligns with actual correctness; a well-calibrated system avoids both overconfidence and underconfidence, which is essential for trustworthy deployment. We assess calibration using Expected Calibration Error (ECE, lower is better), which partitions predictions into $M$ confidence bins and computes the weighted average of the absolute difference between empirical accuracy and predicted confidence.

$$\text{ECE} = \sum_{m=1}^{M} \frac{|B_m|}{n} \left| \text{acc}(B_m) - \text{conf}(B_m) \right|, \tag{1}$$

where $|B_m|$ is the number of predictions in bin $m$, $n$ is the total number of predictions, $\text{acc}(B_m)$ is the empirical accuracy in bin $m$, and $\text{conf}(B_m)$ is the average confidence in bin $m$.

Discriminability, in contrast, assesses whether a model can separate correct from incorrect answers, independent of absolute calibration. High discriminability is particularly important for applications such as ranking or selective prediction. We quantify it using the Area Under the ROC Curve (AUROC, higher is better). Since modern LLMs can explicitly abstain (e.g., "I don't know the answer."), such rejections naturally reflect a form of discrimination. Accordingly, we report AUROC by treating abstentions as confidence = 0 (AUROC Incl.) and ECE by excluding them (ECE Excl.).

---

[2]We use a subset of PopQA with 1,000 randomly sampled questions

To enable comparison with prior work, we evaluate several baseline methods. For all models, we include verbalized numerical confidence (VNC) (Xiong et al., 2023) and semantic uncertainty (SU) (Kossen et al., 2024b). For linguistic confidence, we use a vanilla prompt (Appendix C.1), which directly asks the question and is denoted as LC, and follow Yona et al. (Yona et al., 2024) by additionally instructing models to hedge when uncertain, denoted as LC+ (Appendix C.3). For open-source models, we further consider the self-evaluation method P(True) (Kadavath et al., 2022) (Appendix D.2) and perplexity (Appendix D.3). The implementations of QA, confidence extraction, and grading pipelines are detailed in Appendices C, D, and E, respectively.

We conduct our study on a wide range of modern LLMs, covering both open-source and closed-source models, as well as different model scales. On the smaller end, we include models such as gpt-5-nano and gpt-5-mini, which allow us to examine whether lightweight models can produce reliable confidence estimates. On the larger end, we evaluate powerful models including qwen3-235b-a22b and gpt-5, which represent the frontier of current open- and closed-source systems. We also incorporate intermediate-scale and open-source models such as gpt-oss-20b, gpt-oss-120b, and Llama-4 variants, as well as closed-source alternatives like Claude-3.5 and Claude-sonnet-4. This broad coverage enables us to compare uncertainty estimation methods across diverse architectures, training paradigms, and parameter scales, thereby ensuring that our conclusions are not tied to a specific model family. The implementation details of each model are included in Appendix F.

## 5.2 Confidence Estimation across Models and Datasets

| Models | Method | SimpleQA | | NQ-Open | | PopQA | |
|---|---|---|---|---|---|---|---|
| | | **ECE** | **AUROC** | **ECE** | **AUROC** | **ECE** | **AUROC** |
| gpt-5-mini | LC | 0.4841 | 0.7591 | 0.2543 | 0.5431 | 0.2919 | 0.5666 |
| | LC+ | 0.3420 | **0.8083** | 0.2108 | 0.6019 | 0.2306 | 0.6650 |
| | VNC | 0.2889 | 0.8080 | 0.3223 | **0.7389** | 0.3274 | 0.7913 |
| | SU | **0.2601** | 0.7261 | **0.1332** | 0.7083 | **0.1121** | **0.8078** |
| gpt-oss-120b | LC | 0.6628 | 0.4743 | 0.2692 | 0.6339 | 0.2727 | 0.6792 |
| | LC+ | 0.6304 | 0.5617 | 0.2360 | 0.6844 | 0.2496 | 0.7153 |
| | VNC | 0.5574 | 0.7338 | 0.3117 | 0.7110 | 0.3055 | 0.7393 |
| | P(True) | 0.5778 | 0.4929 | 0.3510 | 0.5226 | 0.3414 | 0.5633 |
| | Perplexity | 0.6906 | 0.7124 | 0.3393 | **0.7178** | 0.3235 | **0.8106** |
| | SU | **0.3448** | **0.7797** | **0.2138** | 0.6795 | **0.2190** | 0.7750 |
| qwen3-235b-a22b | LC | 0.2973 | 0.5102 | 0.2979 | 0.5297 | 0.3263 | 0.5699 |
| | LC+ | **0.1926** | 0.6970 | **0.2072** | 0.6182 | **0.2084** | 0.7080 |
| | VNC | 0.4575 | 0.5267 | 0.4554 | 0.6320 | 0.4968 | 0.6556 |
| | P(True) | 0.3192 | 0.6045 | 0.3639 | 0.6555 | 0.2953 | 0.5356 |
| | Perplexity | 0.4336 | **0.7829** | 0.4326 | **0.6804** | 0.4663 | **0.7801** |
| | SU | 0.3042 | 0.7499 | 0.3337 | 0.6659 | 0.3710 | 0.7288 |

Table 2: Confidence estimation performance across models and datasets, evaluated with ECE↓ (Excl.) and AUROC↑ (Incl.). Best results are in **bold**, second-best are underlined. **LC** denotes linguistic confidence under the vanilla QA prompt (Appendix C.1), while **LC+** augments it with explicit instructions to hedge when uncertain (Appendix C.3). Both LC and LC+ are scored using our proposed mapper, which converts hedging expressions into calibrated confidence scores at near-zero cost. The results highlights that although vanilla LC is poorly calibrated, LC+ achieves competitive calibration and discriminability compared to strong baselines such as semantic uncertainty.

Table 2 reports the confidence estimation performance across datasets and models, with the best results highlighted in bold and the second-best underlined. LC generally performs poorly on both calibration and discriminability. Its AUROC is only slightly above 50%, indicating that responses under the vanilla prompt C.1 can hardly distinguish what the model knows from what it does not. However, after introducing the LC+ prompt C.3, which explicitly instructs the model to hedge when uncertain, LC+ achieves substantially better uncertainty expression performance. Its results are comparable to popular baselines such as semantic uncertainty, suggesting that LLMs are aware of their uncertainty but require explicit prompting to express it through hedging. Notably, LC+ achieves

| Models | ECE (Excl.) | | | | AUROC (Incl.) | | | |
|---|---|---|---|---|---|---|---|---|
| | LC | LC+ | VNC | SU | LC | LC+ | VNC | SU |
| llama-4-maverick | 0.5779 | **0.4305** | 0.6804 | 0.4974 | 0.4970 | 0.5728 | 0.5599 | **0.7305** |
| llama-4-scout | 0.7170 | **0.5016** | 0.8065 | 0.6032 | 0.4991 | 0.5979 | 0.6381 | **0.6804** |
| claude-3-5-haiku | 0.7003 | 0.4343 | 0.7218 | **0.3346** | 0.5452 | **0.7140** | 0.6402 | 0.6972 |
| gpt-oss-20b | 0.7272 | 0.5418 | 0.6869 | **0.2153** | 0.5038 | 0.6593 | 0.7075 | **0.8197** |
| gpt-5-nano | 0.6140 | **0.3260** | 0.3559 | 0.3730 | 0.6944 | 0.7974 | **0.8423** | 0.6743 |
| gpt-5 | 0.4712 | 0.4673 | **0.4312** | – | 0.5043 | 0.5280 | **0.7227** | – |
| claude-sonnet-4 | 0.5418 | **0.1957** | 0.4947 | – | 0.6160 | **0.8179** | 0.4922 | – |

Table 3: ECE↓ (Excl.) and AUROC↑ (Incl.) on SimpleQA across different models. LC and LC+ results are computed using our mapper, with LC+ further prompted to hedge when uncertain. Missing values are denoted by "–".

the best calibration performance on Qwen3-235b-a22b across datasets. Among other methods, SU attains the best overall performance on average.

Additional results on multiple models for SimpleQA are provided in Table 3. Many of the strongest calibration and discriminability results are also observed under LC+. The average calibration and discriminability performance across models on SimpleQA is shown in Figure 4. LC+ shows uncertainty estimation performance close to SU in overall quality and outperform VNC.

| Setting | Method | ACC | ECE | AUROC |
|---|---|---|---|---|
| W/O Reasoning | LC | 0.3008 | 0.4712 | 0.5043 |
| | LC+ | 0.3060 | 0.4673 | 0.5280 |
| | VNC | 0.3439 | 0.4312 | 0.7227 |
| Reasoning | LC | 0.4994 | 0.2780 | 0.5565 |
| | LC+ | 0.4932 | 0.2601 | 0.6204 |
| | VNC | 0.4965 | 0.1979 | 0.8286 |

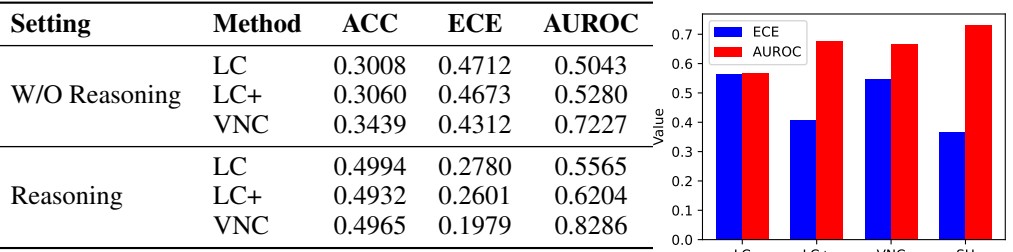

Figure 4: **Left:** Performance of GPT-5 on the SimpleQA dataset under reasoning vs. without reasoning. **Right:** Average ECE and AUROC over multiple models on SimpleQA.

## 5.3 Impact of Reasoning

One of the key features of modern LLMs is their reasoning capability. Prior work has shown that engaging a model's reasoning process by allocating a higher reasoning budget or encouraging step-by-step generation can improve both prediction accuracy and uncertainty estimation, as observed in verbalized numerical confidence (VNC) (Xiong et al., 2023). We further examine its impact on LC using GPT-5 by varying the reasoning effort between two settings: minimal, corresponding to the lowest reasoning budget with 0 additional reasoning tokens per question, and moderate, the default setting that produces roughly 1,000 additional reasoning tokens per question. The results in Table 4 are consistent with prior findings on VNC (Xiong et al., 2023): reasoning improves numerical confidence calibration, and we find that it also substantially enhances linguistic confidence under both LC and LC+. This suggests that reasoning helps the model generate more accurate hedging expressions to convey uncertainty.

## 6 Improve Linguistic Confidence

While explicit prompting can improve uncertainty expression, it requires additional effort and deviates from natural human interaction. A more desirable practice is for models to default to expressing linguistic confidence when directly answering questions (e.g., using the vanilla prompt in Appendix C.1). To test this feasibility, we introduce a fine-tuning framework (Figure 5) that encourages models to express uncertainty more faithfully. Leveraging the strong performance of semantic uncertainty (Farquhar et al., 2024), we adopt it as a proxy ground truth. The framework proceeds as follows: (1) compute semantic uncertainty for the base model and map it into five discrete levels (completely uncertain, lowest, low, moderate, and high); (2) generate uncertain responses at

these levels using the SOTA LLMs such as GPT-5; (3) with the generated sentences, construct question–answer pairs to form the fine-tuning dataset; and (4) fine-tune the base model with LoRA (Hu et al., 2021). We compare the fine-tuned model with multiple methods on SimpleQA and NQ-Open.

## 6.1 TRAINING FRAMEWORK

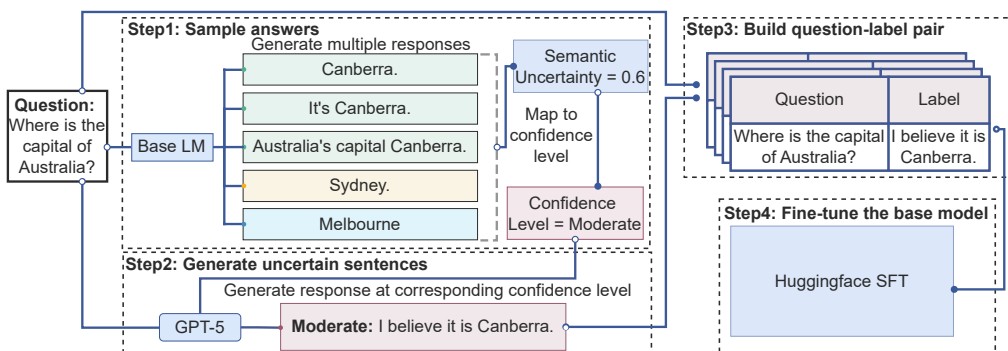

Figure 5: Training framework. **Step 1:** Sample multiple responses for each question using the base model to be trained, then compute its semantic uncertainty and map it to a discrete confidence level. **Step 2:** Use a SOTA LLM (e.g., GPT-5, Grok) to generate sentences at the corresponding confidence level. **Step 3:** Align each question with generated sentences to construct question–label pairs. **Step 4:** Fine-tune the base model on this dataset using Huggingface SFT with LoRA.

**Step 1: Compute semantic uncertainty and map it to a confidence level.** The first step of supervised fine-tuning is to identify a suitable supervision signal. Leveraging the strong performance of semantic uncertainty, we adopt it as a ground truth proxy for a model's uncertainty on a given question. Concretely, we generate 10 answers to each question using the base model and compute semantic uncertainty as the supervision signal. This value is then discretized into five levels (completely uncertain, lowest, low, moderate, and high) via a predefined mapping rule. The mapping rule is included in Table 7 in the Appendix.

**Step 2: Generate uncertain sentences.** Supervised fine-tuning requires natural language sentences as supervision targets. We therefore prompt state-of-the-art commercial LLMs (e.g., GPT-5) to generate uncertain sentences aligned with the designated confidence levels. To streamline implementation, we pre-construct an uncertainty sentence database covering five discrete levels, from which sentences are retrieved as needed. Each training pair consists of a question and a generated uncertain sentence, with the latter serving as the supervision target. The full generation prompt is provided in Appendix B.1. To reduce model-specific bias and enhance variance, we employ multiple SOTA LLMs, each generating 10 responses per confidence level for every question.

**Step 3: Construct fine-tuning dataset.** We randomly sample 200 questions from SimpleQA and, for each question, retrieve 40 LLM-generated responses from Step 2. This results in $200 \times 40 = 8,000$ question–answer pairs in total. The selected 200 questions are excluded from the evaluation set to avoid data leakage.

**Step 4: Fine-tune the base model.** We adopt LoRA (Hu et al., 2021) for efficient fine-tuning, which introduces a small number of trainable low-rank matrices while keeping most pretrained weights frozen, thereby reducing memory and computation costs significantly. Since our objective is to enable the base model to express its internal uncertainty rather than select preferred uncertain tones, we thus choose SFT rather than DPO (Rafailov et al., 2023).

## 6.2 EXPERIMENTAL SETUP

To evaluate the effectiveness of our training framework, we fine-tuned the Qwen3-8B Yang et al. (2025) model. To construct diverse supervision targets and reduce model-specific bias, we leveraged multiple state-of-the-art LLMs, including GPT-5, Grok-3, Claude-Sonnet-4, and Gemini-2.5-Pro; the detailed prompt template is provided in Appendix B.1. Semantic uncertainty was computed

following the standard practice in (Kossen et al., 2024a) using DeBERTa (He et al., 2020) as the entailment model. Fine-tuning was performed with LoRA (rank 32, scaling factor $\alpha = 32$, dropout probability 0.05) with 3 epochs. All experiments were conducted on two Nvidia RTX 3090 GPUs with the base model loaded in 8-bit precision. Since 200 questions from SimpleQA were used for supervision target generation, they were excluded from all subsequent evaluations. To further assess generalization, we also evaluated the fine-tuned model on NQ-Open.

| Models | SimpleQA | | NQ-Open | |
|---|---|---|---|---|
| | ECE | AUROC | ECE | AUROC |
| VNC (Xiong et al., 2023) | 0.8073 | 0.6344 | 0.6591 | 0.5691 |
| SU (Farquhar et al., 2024) | **0.4858** | **0.7489** | 0.3878 | 0.7252 |
| P(True) (Kadavath et al., 2022) | 0.7571 | 0.5586 | 0.4765 | 0.7286 |
| Perplexity (Malinin & Gales, 2020) | 0.8481 | 0.7441 | 0.5846 | 0.7290 |
| LC C.1 | 0.7781 | 0.4790 | 0.4396 | 0.5006 |
| LC+ C.3 | 0.7150 | 0.6628 | 0.4069 | 0.5653 |
| LC (SFT) | 0.7550 | 0.5769 | **0.2837** | **0.7331** |

Table 4: Confidence estimation performance of Qwen3-8B across two QA benchmarks, reported with ECE↓ (Excl.) and AUROC↑ (Incl.). Compared with the base model (LC) and its fine-tuned variant (LC (SFT)), our framework consistently improves both calibration and discriminability, demonstrating its effectiveness. Notably, the fine-tuned model surpasses strong baselines such as verbalized numerical uncertainty (VNC) and semantic uncertainty (SU) on NQ-Open.

### 6.3 CONFIDENCE ESTIMATION PERFORMANCE

We report the confidence estimation performance across confidence estimation methods and datasets in Table 4. Compared with the base model (LC) and the fine-tuned Qwen3-8B (LC (SFT)), our framework achieves consistent improvements in both calibration and discriminability, demonstrating its effectiveness. Notably, the fine-tuned model outperforms strong baseline methods such as verbalized numerical uncertainty and semantic uncertainty on NQ-Open, offering a more user-friendly as well as cost-effective and time-efficient solution. These improvements highlight the substantial potential of expressing uncertainty through hedging language and call for greater attention to this direction in future research. We further provide qualitative examples in the Appendix I.2 showing that model after fine-tuning expresses more reliable linguistic confidence.

## 7 LIMITATION AND DISCUSSION

In this work, we treat the average confidence annotation as ground truth. This choice is suboptimal, as the perception of uncertainty in hedging language varies substantially across individuals and is inherently subjective. For example, for the sentence "I believe the answer is apple," both 30% and 70% confidence assignments could be considered reasonable. Rather than relying on a simple average, modeling hedging uncertainty as a distribution may provide a more faithful representation. Wang et al. (2024), for instance, learned uncertainty distributions for 16 hedging terms. Nevertheless, predicting distributions for diverse hedging expressions remains a nontrivial challenge, and developing models that can capture and predict such distributions is a promising research direction.

Another important avenue is to extend LC beyond QA to reasoning and multimodal settings. In reasoning tasks, uncertainty should be evaluated not only in final answers but also in intermediate steps. In multimodal scenarios, LC may also be conveyed through other modalities—for example, prosodic cues in speech such as intonation, pitch, and hesitation often implicitly signal confidence or uncertainty. Yet many of these natural expressions remain uncaptured by current approaches, underscoring the need for further research.

Our findings demonstrate that LC offers a natural and efficient means for LLMs to communicate uncertainty without requiring access to logits or costly sampling. However, LC is not inherently reliable: under default prompting, most models produce poorly calibrated hedges, and improvements rely on explicit prompting or supervised fine-tuning. This duality highlights both the promise of LC as a human-centered approach to uncertainty communication and its current limitations. Advancing this line of research will require richer datasets, more nuanced mappings, and broader evaluations across tasks and modalities.

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

## A  THE USE OF LARGE LANGUAGE MODELS

We used LLMs for three auxiliary purposes: (i) language refinement (grammar, style, and clarity) during manuscript preparation, (ii) generating candidate uncertain expressions later annotated by humans to build our dataset, and (iii) benchmarking multiple LLMs on linguistic confidence. All scientific ideas, experimental designs, and analyses are the authors' own.

## B  DATASET BUILDING DETAILS

### B.1  PROMPT EXAMPLE TO GENERATE UNCERTAIN EXPRESSIONS

---
**Prompt to generate uncertain expressions**

You are given a question and its ground-truth answer. Your task is to generate 50 answer sentences that express the same answer using different levels of confidence:

10 with high confidence
10 with moderate confidence
10 with low confidence
10 with lowest confidence
10 with complete uncertainty, reject to reply

The wording should vary across the levels, but all responses should convey the same core answer. Focus on natural and diverse expressions of confidence.

Question: At what age was Ken Noda invited by President Ronald Reagan and First Lady Nancy Reagan to perform at the White House?
Answer: 20

---

### B.2  UNCERTAIN EXPRESSION RESPONSE EXAMPLE FROM LLMS

---
**Uncertain expression response example from LLMs**

\*\*High confidence:\*\*

1. Ken Noda was invited to perform at the White House at the age of 20.
2. He was 20 years old when President Ronald Reagan and First Lady Nancy Reagan extended the invitation.
3. At the age of 20, Ken Noda performed at the White House upon invitation.
4. Ken Noda received the White House invitation when he was exactly 20.
5. The age at which Ken Noda was invited to perform at the White House was 20.
6. He got the invitation to perform from the Reagans at age 20.
7. Ken Noda was 20 when he played at the White House at the Reagans' request.
8. It was at 20 years old that Ken Noda was asked to perform at the White House.
9. When Ken Noda was 20, he was invited by President Reagan and First Lady Nancy Reagan to perform.
10. He was precisely 20 years of age when he performed at the White House for the Reagans.

\*\*Moderate confidence:\*\*

11. Ken Noda was likely 20 when he was invited to perform at the White House.
12. I believe Ken Noda was invited by the Reagans to perform when he was 20.
13. He appears to have been 20 years old at the time of the White House performance.
14. If I recall correctly, Ken Noda was 20 when he was asked to perform for the Reagans.
15. It seems that Ken Noda was about 20 when he received the invitation.
16. As far as I know, Ken Noda was 20 during his White House appearance.
17. Ken Noda's age at the time of the invitation was reportedly 20. 18. From what I gather, Ken Noda was 20 when he performed at the White House.
19. My understanding is that Ken Noda was 20 when he was invited.
20. I'm pretty sure Ken Noda was 20 when he got the White House invitation.

\*\*Low confidence:\*\*

21. I think Ken Noda was around 20 when he got invited to the White House.
22. If I'm not mistaken, Ken Noda was about 20 at the time.
23. I have the impression he was 20 years old when he performed at the White House.
24. I believe Ken Noda might have been 20 when he received the invitation.

---

25. I'd guess that Ken Noda was 20 when he was asked to perform for the Reagans.
26. It's possible he was 20 at the time.
27. I think he was probably 20 when the Reagans invited him.
28. If memory serves, Ken Noda would have been 20.
29. I'm under the impression his age was 20 at the time.
30. My best guess is that Ken Noda was 20 years old when he got the invitation.

**Lowest confidence:**

31. I'm not sure, but Ken Noda could have been 20 when he was invited.
32. There's a chance Ken Noda was about 20 at the time.
33. Possibly, his age was 20 when he performed for the Reagans.
34. Maybe Ken Noda was 20 years old at that event.
35. As far as I can tell, he might have been 20.
36. I'm not certain, but it might have been when he was 20.
37. It looks like he may have been 20 when invited to the White House.
38. Perhaps Ken Noda was 20, but I can't say for sure.
39. I have a vague notion that Ken Noda was around 20.
40. My recollection is unclear, but 20 might be the age.

**Complete uncertainty / rejection:**

41. I'm sorry, but I don't know Ken Noda's exact age at the time.
42. I'm unable to provide the specific age at which Ken Noda received the invitation.
43. I don't have sufficient information to answer that question.
44. I can't say for certain what age Ken Noda was when invited.
45. Unfortunately, I don't know the answer to that.
46. I don't have details about Ken Noda's age regarding the White House invitation.
47. Sorry, I'm not sure when Ken Noda was invited to perform.
48. I don't have that information, so I'm unable to respond accurately.
49. I'm afraid I don't know Ken Noda's age at the time of his White House invitation.
50. I don't have access to information confirming Ken Noda's age for that event.

# C  QUESTION-ANSWER (QA) PROMPTS

## C.1  VANILLA PROMPT

> **Vanilla Prompt**
>
> Answer the following question using a succinct (at most one sentence) and full answer.
>
> Question: {question}
> Answer:

## C.2  VANILLA PROMPT FOR VERBALIZED NUMERICAL CONFIDENCE

> **Vanilla Prompt for Verbalized Numerical Confidence**
>
> Answer the following question using a succinct (at most one sentence) and full answer, here is the question: {question}
> Please provide a confidence score between 0 and 100 at the end of your answer in the following JSON format:
> {{
> "answer": Your answer here,
> "confidence_score": number
> }}

## C.3  LC+ PROMPT

> **LC+ Prompt**
>
> Answer the following question using a succinct (at most one sentence) and full answer. If you are uncertain about your answer to the question, convey this uncertainty linguistically by precisely hedging this answer.
>
> Question: {question}
> Answer:

# D CONFIDENCE EXTRACTION IMPLEMENTATION DETAILS

## D.1 VERBALIZED NUMERICAL CONFIDENCE (VNC) EXTRACTION

VNC is generated through direct prompting and extracted by regex.

---
**Regex for VNC Extraction**

```
r'"confidence_score"\s*:\s*([0-9]*\.?[0-9]+(?:[eE][-+]?[0-9]+)?)',
r'confidence_score\s*:\s*([0-9]*\.?[0-9]+(?:[eE][-+]?[0-9]+)?)',
```
---

## D.2 P(TRUE) CONFIDENCE EXTRACTION

---
**P(True) Self-Evaluation Prompt**

Question: {question}
Proposed Answer: {proposed_ answer}
Is the proposed answer:
(A) True
(B) False
The proposed answer is:

Output either A or B with no other text around it.

---

Based on the model response to the self-evaluation prompt above, P(True) is calculated as follows:

$$P(\text{True}) = \exp(-\log(P(\text{True Token})))$$

True-False token probability normalization is omitted, as the observed subsequent top-k token probabilities are negligible.

## D.3 PERPLEXITY CONFIDENCE EXTRACTION

Let $\mathcal{V}$ denote the vocabulary, and $\mathbf{x} \in \mathcal{V}^l$ be an input prompt of $l$ tokens. The model generates a response $\mathbf{y} = (y_1, \ldots, y_n) \in \mathcal{V}^n$, where $n$ is the number of tokens in the generated response. The autoregressive log-likelihood is given by:

$$\log p_\phi(\mathbf{y} \mid \mathbf{x}) = \sum_{i=1}^{n} \log p_\phi(y_i \mid \mathbf{y}_{<i}, \mathbf{x}), \quad \text{where } \mathbf{y}_{<i} = (y_1, \ldots, y_{i-1}). \tag{2}$$

Considering sequence length normalization, the framework exponentiates the average log-likelihood to obtain perplexity:

$$\text{Perplexity} = \exp(-\frac{1}{n} \sum_{i=1}^{n} \log p_\phi(y_i \mid \mathbf{y}_{<i}, \mathbf{x})). \tag{3}$$

## D.4 DIRECT PROMPT TO MAP LINGUISTIC CONFIDENCE

---
**Direct Prompt to map linguistic confidence**

Please provide only a confidence score between 0 and 100, based solely on the degree of confidence expressed in the tone of the following sentence (without using any external or prior knowledge): {sentence}

---

## D.5 PROMPT TO EXTRACT NUMERICAL CONFIDENCE SCORE

---

**Prompt to extract numerical confidence score**

Please provide only a confidence score between 0 and 100, based solely on the degree of confidence expressed in the tone of the following sentence (without using any external or prior knowledge): {sentence}

---

## D.6 DECISIVENSS LLM SCORING PROMPT

---

**Decisiveness Score LLM Judge Prompt**

You will be given a question and a proposed answer. Your task is to extract any assertions made by the proposed answer; for each assertion, you should output a decisiveness score: a number in [0,1] representing how likely you think this assertion is to be true, judged solely based on the provided answer. You should pay special attention to the usage of any hedging modifiers in the original answer, used to convey uncertainty in the truthfulness of the assertion. If the proposed answer punts the question, the extracted assertion should be the empty string and the decisiveness score should be 1.0.

Here are some examples:

Question: criminal minds episode where jj becomes a profiler
Proposed answer: I think JJ officially becomes a profiler in Criminal Minds season 6, episode 2.
Extracted assertion: JJ officially becomes a profiler in Criminal Minds season 6, episode 2.
Decisiveness score: 0.8.

Question: criminal minds episode where jj becomes a profiler
Proposed answer: JJ officially becomes a profiler in Criminal Minds season 6, in either the first or second episode.
Extracted assertion: JJ officially becomes a profiler in Criminal Minds season 6, episode 1.
Decisiveness score: 0.5.
Extracted assertion: JJ officially becomes a profiler in Criminal Minds season 6, episode 2.
Decisiveness score: 0.5.

Question: criminal minds episode where jj becomes a profiler
Proposed answer: I'm not really sure about this, but I think the episode in which JJ officially becomes a profiler in Criminal Minds may be episode 2 in season 6.
Extracted assertion: JJ officially becomes a profiler in Criminal Minds season 6, episode 2.
Decisiveness score: 0.6.

Question: criminal minds episode where jj becomes a profiler
Proposed answer: I don't know which episode you're referring to.
Extracted assertion:
Decisiveness score: 1.0
Question: {question}

Proposed answer: {response}

---

# E    GRADER PROMPTING DETAILS

**Grading SimpleQA and NQ-Open**    We employ GPT-5-mini as the grader model. We only consider the "best answer" in the NQ-Open dataset due to the LLM grader's capability to infer answers with equivalent meanings. If the grader fails to grade or output as instructed, the question is automatically classified as "NOT_ATTEMPTED".

---

**SimpleQA and NQ-Open Grader Prompt**

Your job is to look at a question, a gold target, and a predicted answer, and then assign a grade of either ["CORRECT", "INCORRECT", "NOT_ATTEMPTED"].
First, I will give examples of each grade, and then you will grade a new example.

The following are examples of CORRECT predicted answers.
```
Question: What are the names of Barack Obama's children?
Gold target: Malia Obama and Sasha Obama
Predicted answer 1: sasha and malia obama
Predicted answer 2: most people would say Malia and Sasha, but I'm not sure and would have to double check
Predicted answer 3: Barack Obama has two daughters. Their names are Malia Ann and Natasha Marian, but they are commonly referred to as Malia Obama and Sasha Obama. Malia was born on July 4, 1998, and Sasha was born on June 10, 2001.
```
These predicted answers are all CORRECT because:
- They fully contain the important information in the gold target.
- They do not contain any information that contradicts the gold target.
- Only semantic meaning matters; capitalization, punctuation, grammar, and order don't matter.
- Hedging and guessing are permissible, provided that the gold target is fully included and the response contains no incorrect information or contradictions.

The following are examples of INCORRECT predicted answers.
```
Question: What are the names of Barack Obama's children?
Gold target: Malia and Sasha
Predicted answer 1: Malia.
Predicted answer 2: Malia, Sasha, and Susan.
Predicted answer 3: Barack Obama does not have any children.
Predicted answer 4: I think it's either Malia and Sasha. Or it could be Malia and Jackie. Or it could be Joey and Malia.
Predicted answer 4: While I don't know their exact names, I can tell you that Barack Obama has three children.
Predicted answer 5: It's possible you may mean Betsy and Olivia. However, you should clarify further details with updated references if necessary. Is that the correct answer?
Predicted answer 6: It may be the case that Obama's child is named James. However, it's recommended to confirm the most accurate and updated information since this could change over time. This model may not always reflect the most current information.
```
These predicted answers are all INCORRECT because:
- A factual statement in the answer contradicts the gold target. Incorrect statements that have some hedging (e.g., "it is possible that", "although i'm not sure, i think") are also considered incorrect.

The following are examples of NOT_ATTEMPTED predicted answers.
```
Question: What are the names of Barack Obama's children?
Gold target: Malia and Sasha
Predicted answer 1: I don't know.
Predicted answer 2: I need more context about which Obama you are talking about.
Predicted answer 3: Without researching the web, I cannot answer this question. However, I can tell you that Barack Obama has two children.
Predicted answer 4: Barack Obama has two children. I know that one of them is Malia, but I'm not sure about the other one.
```
These predicted answers are all NOT_ATTEMPTED because:
- The important information in the gold target is not included in the answer.
- No statements in the answer contradict the gold target.

Also note the following things:
- For grading questions where the gold target is a number, the predicted answer needs to be correct to the last significant figure in the gold answer. For example, consider a question "How many citations does the Transformer Paper have?" with gold target "120k".
- Predicted answers "120k", "124k", and 115k" are all CORRECT.
- Predicted answers "100k" and "113k" are INCORRECT.
- Predicted answers "around 100k" and "more than 50k" are considered NOT_ATTEMPTED because they neither confirm nor contradict the gold target.
- The gold target may contain more information than the question. In such cases, the predicted answer only needs to contain the information that is in the question.
- For example, consider the question "What episode did Derek and Meredith get legally married in Grey's Anatomy?" with gold target "Season 7, Episode 20: White Wedding". Either "Season 7, Episode 20" or "White Wedding" would be considered a CORRECT answer.

---

- Do not punish predicted answers if they omit information that would be clearly inferred from the question.
- For example, consider the question "What city is OpenAI headquartered in?" and the gold target "San Francisco, California". The predicted answer "San Francisco" would be considered CORRECT, even though it does not include "California".
- Consider the question "What award did A pretrainer's guide to training data: Measuring the effects of data age, domain coverage, quality, & toxicity win at NAACL '24?", the gold target is "Outstanding Paper Award". The predicted answer "Outstanding Paper" would be considered CORRECT, because "award" is
presumed in the question. - For the question "What is the height of Jason Wei in meters?", the gold target is "1.73 m". The predicted answer "1.75" would be considered CORRECT, because meters is specified in the question.
- For the question "What is the name of Barack Obama's wife?", the gold target is "Michelle Obama". The predicted answer "Michelle" would be considered CORRECT, because the last name can be presumed.
- Do not punish for typos in people's name if it's clearly the same name.
- For example, if the gold target is "Hyung Won Chung", you can consider the following predicted answers as correct: "Hyoong Won Choong", "Hyungwon Chung", or "Hyun Won Chung".

Here is a new example. Simply reply with either CORRECT, INCORRECT, NOT ATTEMPTED. Don't apologize or correct yourself if there was a mistake; we are just trying to grade the answer.
```
Question: {question}
Gold target: {target}
Predicted answer: {predicted_answer}
```

Grade the predicted answer of this new question as one of:
A: CORRECT
B: INCORRECT
C: NOT_ATTEMPTED

Just return the letters "A", "B", or "C", with no text around it.

**Grading PopQA** We employ GPT-5-mini as the grader model. The grader prompt is a slight variation from SimpleQA's to adapt the list-structured answers in the PopQA dataset. If the grader fails to grade or output as instructed, the question is automatically classified as "NOT_ATTEMPTED".

### PopQA Grader Prompt

Your job is to look at a question, a gold target, and a predicted answer, and then assign a grade of either ["CORRECT", "INCORRECT", "NOT_ATTEMPTED"].
A gold target is a list of accepted answers. If any of the accepted answers agrees with the predicted answer, the grade will be "CORRECT".
First, I will give examples of each grade, and then you will grade a new example.

The following are examples of CORRECT predicted answers.
```
Question: What are the names of Barack Obama's children?
Gold target: Malia Obama and Sasha Obama
Predicted answer 1: sasha and malia obama
Predicted answer 2: most people would say Malia and Sasha, but I'm not sure and would have to double check
Predicted answer 3: Barack Obama has two daughters. Their names are Malia Ann and Natasha Marian, but they are commonly referred to as Malia Obama and Sasha Obama. Malia was born on July 4, 1998, and Sasha was born on June 10, 2001.
```
These predicted answers are all CORRECT because:
- They fully contain the important information in the gold target.
- They do not contain any information that contradicts the gold target.
- Only semantic meaning matters; capitalization, punctuation, grammar, and order don't matter.
- Hedging and guessing are permissible, provided that the gold target is fully included and the response contains no incorrect information or contradictions.

The following are examples of INCORRECT predicted answers.
```
Question: What are the names of Barack Obama's children?
Gold target: Malia and Sasha
Predicted answer 1: Malia.
Predicted answer 2: Malia, Sasha, and Susan.
Predicted answer 3: Barack Obama does not have any children.
Predicted answer 4: I think it's either Malia and Sasha. Or it could be Malia and Jackie. Or it could be Joey and Malia.
Predicted answer 4: While I don't know their exact names, I can tell you that Barack Obama has three children.
Predicted answer 5: It's possible you may mean Betsy and Olivia. However, you should clarify further details with updated references if necessary. Is that the correct answer?
Predicted answer 6: It may be the case that Obama's child is named James. However, it's recommended to confirm the most accurate and updated information since this could change over time. This model may not always reflect the most current information.
```
These predicted answers are all INCORRECT because:

- A factual statement in the answer contradicts the gold target. Incorrect statements that have some hedging (e.g., "it is possible that", "although i'm not sure, i think") are also considered incorrect.

The following are examples of NOT_ATTEMPTED predicted answers.
```
Question: What are the names of Barack Obama's children?
Gold target: Malia and Sasha
Predicted answer 1: I don't know.
Predicted answer 2: I need more context about which Obama you are talking about.
Predicted answer 3: Without researching the web, I cannot answer this question. However, I can tell you that Barack Obama has two children.
Predicted answer 4: Barack Obama has two children. I know that one of them is Malia, but I'm not sure about the other one.
```
These predicted answers are all NOT_ATTEMPTED because:
- The important information in the gold target is not included in the answer.
- No statements in the answer contradict the gold target.

Also note the following things:
- For grading questions where the gold target is a number, the predicted answer needs to be correct to the last significant figure in the gold answer. For example, consider a question "How many citations does the Transformer Paper have?" with gold target "120k".
- Predicted answers "120k", "124k", and 115k" are all CORRECT.
- Predicted answers "100k" and "113k" are INCORRECT.
- Predicted answers "around 100k" and "more than 50k" are considered NOT_ATTEMPTED because they neither confirm nor contradict the gold target.
- The gold target may contain more information than the question. In such cases, the predicted answer only needs to contain the information that is in the question.
- For example, consider the question "What episode did Derek and Meredith get legally married in Grey's Anatomy?" with gold target "Season 7, Episode 20: White Wedding". Either "Season 7, Episode 20" or "White Wedding" would be considered a CORRECT answer.
- Do not punish predicted answers if they omit information that would be clearly inferred from the question.
- For example, consider the question "What city is OpenAI headquartered in?" and the gold target "San Francisco, California". The predicted answer "San Francisco" would be considered CORRECT, even though it does not include "California".
- Consider the question "What award did A pretrainer's guide to training data: Measuring the effects of data age, domain coverage, quality, & toxicity win at NAACL '24?", the gold target is "Outstanding Paper Award". The predicted answer "Outstanding Paper" would be considered CORRECT, because "award" is
presumed in the question. - For the question "What is the height of Jason Wei in meters?", the gold target is "1.73 m". The predicted answer "1.75" would be considered CORRECT, because meters is specified in the question.
- For the question "What is the name of Barack Obama's wife?", the gold target is "Michelle Obama". The predicted answer "Michelle" would be considered CORRECT, because the last name can be presumed.
- Do not punish for typos in people's name if it's clearly the same name.
- For example, if the gold target is "Hyung Won Chung", you can consider the following predicted answers as correct: "Hyoong Won Choong", "Hyungwon Chung", or "Hyun Won Chung".

Here is a new example. Simply reply with either CORRECT, INCORRECT, NOT ATTEMPTED. Don't apologize or correct yourself if there was a mistake; we are just trying to grade the answer.
```
Question: {question}
Gold target: {target}
Predicted answer: {predicted_answer}
```

Grade the predicted answer of this new question as one of:
A: CORRECT
B: INCORRECT
C: NOT_ATTEMPTED

Just return the letters "A", "B", or "C", with no text around it.

## F  MODEL SETUP

The details tabulated below outline the model sources and parameters chosen for our evaluation. In particular, we also evaluate GPT-5 at the minimal and medium reasoning levels. Other unstated parameters are left as their system default values.

| Model | Platform | Max Tokens | Reasoning Effort |
|---|---|---|---|
| GPT-5-mini | OpenAI | / | minimal |
| GPT-5-nano | OpenAI | / | minimal |
| GPT-5 | OpenAI | / | minimal vs. moderate |
| Llama-4-Maverick-17B-128E-Instruct-FP8 | Together AI | / | / |
| Llama-4-Scout-17B-16E-Instruct | Together AI | / | / |
| GPT-OSS-20B | Together AI | / | low |
| GPT-OSS-120B | Together AI | / | low |
| Qwen3-235B-A22B-Instruct-2507-tput | Together AI | / | off |
| Claude-Sonnet-4-20250514 | Anthropic | 1000 | off |
| Claude-3-5-Haiku-20241022 | Anthropic | 1000 | off |

# G  BUILD BENCHMARK

## G.1  AMAZON MECHANICAL TURK SURVEY EXAMPLE

**Warning: Each worker may complete a maximum of five assignments. Submissions exceeding this limit will be rejected.**

**Confidence Annotation Task**

You will see **105 question–answer pairs**. Your job is to rate how confident the *answer* sounds, using a slider from **0 (very uncertain)** to **100 (very confident)**.

**Important:** Ignore factual correctness—focus only on word choice and tone.

**Quality check:** 5 of the 105 items are validation pairs with expert ratings. If your scores are reasonably close, your work will be **approved**; otherwise it may be **rejected**.

**Example (with context)**

*Question:* What's the month, day, and year of the online publication of the paper "Articulatory constraints on stop insertion and elision in consonant clusters"?
- "There's **no doubt** the date is 5th September 2011." → **100**
- "I **believe** the paper was published on September 5th 2011." → **60**
- "It **might have** been on April 15 2011." → **30**
- "I'm sorry, but I **can't definitively** answer that." → **0**

[ View instructions ]

**Sentence 1/105:** ${sentence_1}

Low confidence  ○————|——————  High confidence
                     50

**Sentence 2/105:** ${sentence_2}

Low confidence  ○————|——————  High confidence
                     50

**Sentence 3/105:** ${sentence_3}

Low confidence  ○————|——————  High confidence
                     50

**Sentence 4/105:** ${sentence_4}

Low confidence  ○————|——————  High confidence
                     50

Figure 6: Amazon Mechanical Turk survey example

## G.2  BENCHMARK EXAMPLE

| Uncertain Expression | A1 | A2 | A3 | A4 | A5 |
|---|---|---|---|---|---|
| I cannot confirm the specific day, month, and year for that performance. | 30 | 40 | 35 | 25 | 45 |
| If I recall correctly, it might have been 26 October 2021. | 50 | 55 | 60 | 45 | 65 |
| The general consensus is that Cómbita's founding year is 1586. | 70 | 75 | 80 | 65 | 60 |
| I'm positive that the founding year of Pajarito, Boyacá, is 1853. | 90 | 85 | 95 | 88 | 92 |
| The first meeting between Luce and Graham happened in South Carolina. | 80 | 70 | 75 | 85 | 78 |
| I would say Zerxus is 6 feet in height. | 60 | 65 | 55 | 70 | 62 |

Table 5: Examples from our benchmark: uncertain expressions with five human annotations (0–100).

## H  TRAIN A CONFIDENCE MAPPER

| Hedging Language | Example |
|---|---|
| Lexical Verbs | The results *appear* to show… |
| Modal Verbs | The findings *might* provide… |
| Reporting Verbs | Smith (2010) *claims* that… |
| Adverbs of Frequency | Students *often* use smartphones… |
| Adverbs of Probability | This research *probably* indicates… |
| Adjectives of Probability | It is *unlikely* that these findings… |
| Adjectives of Quantity | *Many* students today find English… |
| Modal Nouns | One *assumption* is that… |
| 'It' Phrases | *It can be argued* that… |

Table 6: Examples of hedging language in academic writing.

## I  FINE-TUNING MODEL FOR BETTER LC

### I.1  CONFIDENCE MAPPING RULE

| High | Moderate | Low | Lowest | Completely Uncertain |
|---|---|---|---|---|
| $(0.8, 1.0]$ | $(0.6, 0.8]$ | $(0.4, 0.6]$ | $(0.2, 0.4]$ | $[0, 0.2]$ |

Table 7: Confidence interval mapping used for constructing the supervised fine-tuning dataset.

### I.2  QUALITATIVE EXAMPLES

We randomly sample two examples from confidence estimation datasets and compare LC and LC(SFT). Differences are highlighted and results are shown in Table 8.

| Question | LC (Base) | LC (SFT) |
|---|---|---|
| Which one of the 1990 Fly Girls from the series *In Living Color* stayed for five seasons? | The 1990 Fly Girls from the series *In Living Color* who stayed for five seasons was Rosie Perez. | Deidre Lang is **likely** the one who stayed for five seasons. |
| Who sang the first line of *We Are the World*? | The first line of "We Are the World" was sung by Michael Jackson. | I have some **uncertainty**, but I think the first line of "We Are the World" was sung by Lionel Richie. |

Table 8: Qualitative comparison of LC and LC (SFT) responses to the same questions. Fine-tuning (LC (SFT)) encourages the use of hedging expressions, making uncertainty more explicit.

