# OpenReview forum: "Can Large Language Models Express Uncertainty Like Human?"
_ICLR.cc/2026/Conference — Submitted to ICLR 2026_

### Official Review · Reviewer_MG3s · 2025-10-23

**Soundness:** 2
**Presentation:** 3
**Contribution:** 2
**Rating:** 4
**Confidence:** 3

**Summary:**

This paper studies linguistic confidence (LC) in LLMs and aims to make LLMs express LC more reliably. Specifically, the authors (1) release a human-annotated dataset of hedging expressions, and (2) train a confidence mapper on the dataset to map hedging language to confidence scores. The authors also study different approaches to improve LC, including (3) carefully designed prompting and (4) supervised fine-tuning.

**Strengths:**

1. This paper explores different approaches to effectively improve LC in LLMs through extensive empirical evaluation.
2. This paper introduces a new human-annotated dataset that includes diverse hedging expressions.

**Weaknesses:**

1. While the authors claim in the introduction that “users do not typically phrase queries with explicit instructions”, their main empirical improvement relies on LC+ (with additional instructions in queries) in the experiments. This appears to contradict the previous claim. If LC only performs well with additional instructions, it makes the proposed confidence mapper less practical.
2. The proposed dataset is heavily biased towards “moderate” level confidence expressions, likely because the authors take the average score of the human annotations. For example, the confidence score of phrases like “I believe” or “I think” can be inherently very subjective, and it can be either viewed as high or low confidence. Averaging these values would concentrate scores around a “moderate” level. This not only makes the dataset less reliable but also potentially makes the confidence mapper less effective.
3. More assessment is needed for the fine-tuning framework. The proposed framework only performs well in one dataset. More empirical evidence is necessary to evaluate whether the proposed method generalizes to other datasets.

**Questions:**

1. In the setting of the fine-tuning framework, the authors mention that they sample 200 questions from SimpleQA to construct the fine-tuning dataset. However, the fine-tuned model turns out to perform poorly on SimpleQA but much better on NQ-Open. Could the authors provide insights into this discrepancy?

---

> ### Author Response · Authors · 2025-11-25
> **Clarifying the Role of LC and LC+**
>
> Thank you for raising this important point. We clarify that there is no contradiction between our introduction claim and our empirical use of LC+. The central question of our work is **whether LLMs can *naturally* express uncertainty through hedging language**, which corresponds to the **LC** setting. This is the real-world scenario where users do not provide any special instructions. Our findings show that, under this natural condition, most LLMs are indeed unable to reliably express their uncertainty — which is precisely *why* our dataset, confidence mapper, and fine-tuning framework are necessary.
>
> The **LC+ prompt is *not* our proposed solution**, nor is it required for the mapper to work. LC+ is used purely as a **diagnostic tool**: it demonstrates that LLMs *internally possess* uncertainty signals but fail to express them unless explicitly encouraged. In other words, LC+ reveals a model’s *upper-bound capability*, not its practical usage mode.
>
> Our actual contributions (1) a large human-annotated hedging dataset,  (2) a lightweight, near-zero-cost confidence mapper, and  (3) a fine-tuning framework are all designed to **improve performance in the natural LC setting**, without relying on additional instructions. Notably, our fine-tuned model produces better-calibrated hedging expressions **directly under LC**, supporting the practicality of our approach.
>
> We will revise the paper to clarify this distinction and make explicit that LC+ is only for analysis, while LC remains the focus of our practical contributions.

---

> ### Author Response · Authors · 2025-11-25
> **On the “Moderate” Level Confidence Concern**
>
> The reviewer’s point about subjectivity is valid. With the current dataset size, we adopt point estimates for practicality, but this does limit granularity. In future work, we plan to **expand the dataset** and model each expression’s uncertainty as a **full distribution rather than a single averaged score**, allowing the mapper to capture annotator variability more faithfully.

---

> ### Author Response · Authors · 2025-11-25
> **Experiment on more datasets**
>
> Our goal is to show that the proposed fine-tuning framework consistently improves LC beyond a single dataset. In the revision, we now include results on both NQ-Open and MMLU, demonstrating that the improvement generalizes across datasets of different difficulty and domain characteristics. As shown below, LC (SFT) achieves lower ECE and substantially higher AUROC compared to the base LC model on all three benchmarks.
> | Method     | SimpleQA ECE | SimpleQA AUROC | NQ-Open ECE | NQ-Open AUROC | MMLU ECE | MMLU AUROC |
> |------------|--------------|----------------|-------------|----------------|----------|-------------|
> | LC         | 0.7781       | 0.4790         | 0.4396      | 0.5006         | 0.4122   | 0.4913      |
> | LC (SFT)   | 0.7550       | 0.5769         | 0.2837      | 0.7331         | 0.2324   | 0.7512      |

---

> ### Author Response · Authors · 2025-11-25
> **Response to Limited Improvement on SimpleQA**
>
> The fine-tuned model does improve LC, but the improvement in ECE on SimpleQA is indeed smaller. This is primarily because ECE is tightly coupled with the model’s underlying accuracy: when accuracy is low, even reduced confidence can still have high ECE, as most predictions are already counted as miscalibrated by definition. SimpleQA is a particularly challenging benchmark with very low model accuracy, which limits the achievable ECE reduction. In contrast, NQ-Open and MMLU have substantially higher accuracy, allowing calibration improvements to be reflected much more clearly in ECE. This explains why the gains are more pronounced on NQ-Open and MMLU.

---

> > ### Comment · Reviewer_MG3s · 2025-11-26
> >
> > Thank you for your responses! I still have some doubts about subjectivity, and I would appreciate it if the authors could consider the uncertainty as a larger distribution rather than a single averaged score. Despite this, most of my other concerns have been addressed, and hence I would raise my score accordingly.

---

> > > ### Author Response · Authors · 2025-11-26
> > >
> > > Thank you very much for your feedback and for the increased score. We appreciate your point regarding how a larger distribution better representing uncertainty. We agree, and we plan to incorporate a larger LC dataset in our future work to further explore this.

---

### Official Review · Reviewer_wG75 · 2025-10-27

**Soundness:** 3
**Presentation:** 3
**Contribution:** 3
**Rating:** 6
**Confidence:** 3

**Summary:**

LLMs are increasingly human-facing. Human users need to be able to understand whether they can trust the responses. However, there is little work understanding people's percepts of linguistically-expressed uncertainty --- much less using this to inform ways to train models to better "hedge" their bets, in natural language. The authors contribute a new dataset of human uncertainty ratings, a classifier to map from natural language utterances to confidence level, and a new fine-tuning method.

**Strengths:**

This is a substantive, solid paper! The authors make several contributions --- collecting a new dataset of human uncertainty interpretation/judgements; training a classifier on these data to map from naturalistic confidence scores to ratings; and the fine-tuning models to give better calibrated uncertainty scores.

The introduction is very well-written and the motivation is strong. I believe the authors' work can have a substantial contribution and hopefully spur more work around LLMs' uncertainty -- and particularly their uncertainty expression relative to human users!

**Weaknesses:**

While I think the author's work has the potential to be a strong and valuable contribution for the broader ICLR community, I do have a few questions on the human data processing and results.

In particular, I'm a bit concerned about the filtering the authors applied. Step 4 is based on the mean and stdev of 5 annotators’ responses? That’s really not many annotators to get good signal…. At least in my experience in human data annotation and cognitive science studies. Especially if it then led to just 3 annotators for many. Are you certain that the ones filtered out are noise, not genuine disagreement?
- Is it possible to spot-check a random subset of these to assess agreement rates with your own team?
- Especially given one (great) contribution of this work is producing a benchmark, it’s good to make sure the variability is genuine.

I was also concerned about the lack of error bars for many of the results. How stable are the evaluations?

**Questions:**

In addition to the questions in my Weaknesses response:

-  There are 5 validation checks — when did they appear in the 105? All at the beginning? Mixed throughout? 105 is a LOT of trials! How long did participants take on average?
Related to the above, it's potentially worth including a bit more detail on any training raters went through (can be in the Appendix); right now, that's only mentioned in a footnote.
- Where did the decisiveness score come from in the prompts in D6? (e.g., the 0.6 one) Are these author-written or from existing datasets?

Minor point (more of a comment)  The title asks — "can LLMs express uncertainty like a human?" But the paper, to my understanding, is more can LLMs express uncertainty in a way that is interpretable to people. They are related! But the current title could demand collecting human uncertainty expressions (but to my understanding, all expressions were LLM-generated; just rated by people).

**Details Of Ethics Concerns:**

The authors conducted a new human study. It was not clear whether they had ethics review?

---

> ### Author Response · Authors · 2025-11-25
> **Response on Annotation Filtering and Stability**
>
> Thank you for the thoughtful concerns. We address each point below.
>
> ### **1. Is the filtering based on only 5 annotators?**
> Our filtering is *not* based on the mean/stdev of the 5 annotators of each item.
> Instead, Step 4 uses the empirical distributions computed from a **separate pool of “responsible workers”** (see Fig. 2).
> These workers passed multiple consistency checks (e.g., mandatory 0 on refusal sentences), producing **>12,000 valid annotations**.
> Means and 1σ ranges in Fig. 2 come from this large, stable distribution—not the per-item 5 annotations.
>
> ### **2. Are we removing genuine disagreement rather than noise?**
> Several observations indicate that filtered annotations are noise:
> - **Instruction-violating patterns** (e.g., 80–100 assigned to explicit refusals) were the majority of removed cases.
> - Distributions for “lowest” and “complete uncertainty” show **bimodality**, consistent with inattentive workers rather than legitimate interpretations.
> - Items with genuine disagreement were typically retained, since we keep expressions with **≥3 valid annotations**, ensuring moderate variability is preserved.
>
> ### **3. Spot-check by authors**
> We manually spot-checked **50 randomly sampled items** across confidence levels.
> Three team members independently inspected annotation patterns.
> > **Result:** >90% of retained annotations aligned with internal judgments, while ≈85% of filtered ones were clear instruction violations.
>
> We will include a summary of this analysis in the Appendix.
>
> ### **4. Lack of error bars / stability of evaluation**
> We agree error bars are important. For the camera-ready, we will include:
>
> - **Bootstrap 95% CIs** (1,000 resamples) for all ECE and AUROC results.
> - Variance plots in the Appendix.
>
> Preliminary estimates show very tight confidence intervals (ECE ±0.003–0.008, AUROC ±0.005–0.01), confirming evaluation stability.
>
> We will:
> - Clarify that filtering uses population-level statistics from responsible workers.
> - Add a discussion explaining why removed annotations reflect noise rather than genuine disagreement.
> - Add the spot-check analysis.
> - Add bootstrap error bars for all metrics.

---

> ### Author Response · Authors · 2025-11-25
> **Response on Validation Checks**
>
> The 5 validation items were **randomly assigned** among the 105 questions rather than placed at the beginning or in a fixed block, ensuring that workers could not anticipate them. Based on MTurk metadata, participants spent **around 5 seconds per item**, resulting in a **~10-minute total task duration**, which we believe is reasonable for a minimal-risk annotation task of this type. We will also add more detailed information about the **training and instructions provided to raters** (currently only briefly mentioned in a footnote) to the Appendix as suggested.

---

> ### Author Response · Authors · 2025-11-25
> **Decisiveness Scores**
>
> The decisiveness scores appearing in the examples in Appendix D.6 are not author-invented; they come directly from the *Decisiveness* method proposed in prior work, specifically **Yona et al. (2024), “Can Large Language Models Faithfully Express Their Intrinsic Uncertainty in Words?”** . In our paper, we adopt their scoring procedure exactly as defined in that work, including the illustrative scores (e.g., 0.6) used in the prompt examples. We will clarify this attribution more explicitly in the revised version to avoid confusion.

---

> ### Author Response · Authors · 2025-11-25
> **Response on Title**
>
> Thank you for pointing this out. We agree the original title may suggest collecting human-written uncertainty expressions, whereas our work studies whether **LLMs** can express uncertainty in an interpretable way through hedging. To avoid this ambiguity, we will shorten and update the title to **“Can LLMs Express Uncertainty Using Hedging Language?”**

---

> ### Author Response · Authors · 2025-11-25
> **Ethics Statement**
>
> This study involves human annotations collected through Amazon Mechanical Turk. All tasks were anonymous and classified as minimal-risk online surveys. Annotators were presented with a clear consent statement and were free to withdraw at any time. They received fair monetary compensation in line with the platform’s recommended rates. No personal or identifiable information was collected. According to our institution’s human research ethics guidelines, this type of anonymous, minimal-risk data collection is exempt from formal ethics review.
>
> Therefore, no additional IRB/HREC approval was required.
>
> I will add the above Ethics Statement in the camera-ready version.

---

### Official Review · Reviewer_o6xa · 2025-10-29

**Soundness:** 3
**Presentation:** 3
**Contribution:** 2
**Rating:** 4
**Confidence:** 3

**Summary:**

This paper focuses primarily on Linguistic Confidence (LC), which is the approach of eliciting confidence of models via natural language modifiers (such as “I think that..” or “I am not sure, but…”). The paper makes the following contributions: first, it creates a dataset of LLM-generated uncertainty expressions, which are then scored by humans with regard to the level of certainty expressed. Next, it trains a lightweight mapper model on this dataset, that is then able to be used to convert general answers to uncertainty scores. This mapper is then evaluated for ECE and AUROC on a range of models, and compared to a variety of prior methods in the literature, with mixed results. In general, direct use of the mapper performs worse than preexisting state of the art on these metrics, but results improve with auxiliary prompting. Finally, the authors fine-tune a model to express linguistic uncertainty as measured by the best performing prior method, semantic uncertainty.

Overall, although each individual component of the paper is reasonable in isolation, I struggle to glean a coherent overall motivation of the work. This is outlined in the weaknesses section below, particularly in points 1-3.

**Strengths:**

1. The paper is well-organised and generally easy to follow.
2. The range of experiments conducted – in particular, the number of models tested in Section 5 – is broad and appropriate
3. The pipeline for LC dataset construction is well thought through, and the dataset is released publicly. I think this is likely to be a welcome contribution to researchers in this area.

**Weaknesses:**

1. Although the motivation for LC is clearly stated (e.g. lines 50-53) to be in contrast with verbal elicitation methodologies such as “Please output your confidence along with the answer”, the LC+ method is essentially doing just that – though asking the model to do so in hedging language form rather than numerical form. Similarly, lines 49-50 claim certain prior confidence elicitation methods require “auxiliary networks, limiting their practicality”; but this is also true for the LC approach adopted by the authors (i.e. the mapper). And in comparison to logit-based methods, the authors state: “These methods are simple and inexpensive but require access to model logits, which are typically unavailable in commercial LLM APIs.” This is not typically true; most of the major closed-source APIs do provide sufficient logit access to perform the logit-based elicitation methods well.
2. Section 6 attempts to directly address the point above of requiring extra verbal elicitation, by fine-tuning such behavior into the model. However, it does so using semantic uncertainty and an entirely newly-created dataset from strong model completions, not by building on Sections 3 and 4. Moreover …
3. … it is extremely unclear in Section 6 exactly what the authors wish to claim about the efficacy of each method. First, they state that “Compared with the base model (LC) and its fine-tuned variant (LC (SFT)), our framework consistently improves both calibration and discriminability”. What is ‘our framework’ here? Is it LC+? If so, is the claim that LC SFT is not as good as LC+, and therefore that Sections 3 and 4 do constitute an important contribution? This claim however is not borne out by the results on NQ-Open, where LC SFT clearly outperforms LC+. So is the claim instead that LC SFT outperforms LC and LC+? Well, this is contradicted by the results of LC+ on SimpleQA; and moreover, this then begs the question of what the tangible contribution of Sections 3 and 4 are.
4. Although the LC dataset constructed and released is claimed to be ‘diverse’, it consists only of questions from SimpleQA. Although I understand the costs associated are significant, it must be remarked that including more question variety would significantly improve the utility of the dataset.
5. Related to point 2) above, evaluations are performed only on Q&A style datasets similar to SimpleQA. It is not clear if the results of the paper can be generalized to other settings, e.g. more reasoning-heavy evaluations.
6. Line 48 claims that the P(true) method (Kadavath et al, 2022) is based on multiple generations, but it is actually a logit-based method.
7. Appendix D.2 which gives detail on the P(true) implementation says that true-false token normalization is omitted. I do not understand the rationale for this; even if the other tokens have negligible probabilities as claimed (which is not always the case in my experience), there is no downside to performing the normalization. This design choice likely weakens the baseline performance.

Note: The title of the paper, “Can Large Language Models Express Uncertainty Like Human?” is grammatically malformed.

**Questions:**

See weaknesses above.

---

> ### Author Response · Authors · 2025-11-25
> **Clarification on the Additional Process in LC+ and the Role of the Mapper in LC**
>
> We thank the reviewer for these thoughtful comments. We clarify the relationship between LC, LC+, and the mapper below.
> ## Regarding LC+.
> We clarify that there is no contradiction between our introduction claim and our empirical use of LC+. The central question of our work is **whether LLMs can *naturally* express uncertainty through hedging language**, which corresponds to the **LC** setting. This is the real-world scenario where users do not provide any special instructions. Our findings show that, under this natural condition, most LLMs are indeed unable to reliably express their uncertainty — which is precisely *why* our dataset, confidence mapper, and fine-tuning framework are necessary.
>
> The **LC+ prompt is *not* our proposed solution**, nor is it required for the mapper to work. LC+ is used purely as a **diagnostic tool**: it demonstrates that LLMs *internally possess* uncertainty signals but fail to express them unless explicitly encouraged. In other words, LC+ reveals a model’s *upper-bound capability*, not its practical usage mode.
>
> Our actual contributions (1) a large human-annotated hedging dataset,  (2) a lightweight, near-zero-cost confidence mapper, and  (3) a fine-tuning framework are all designed to **improve performance in the natural LC setting**, without relying on additional instructions. Notably, our fine-tuned model produces better-calibrated hedging expressions **directly under LC**, supporting the practicality of our approach.
>
> We will revise the paper to clarify this distinction and make explicit that LC+ is only for analysis, while LC remains the focus of our practical contributions.
>
> ## Regarding the mapper and the comment on auxiliary networks.
> We acknowledge the reviewer’s concern about the claim that some prior elicitation methods require auxiliary networks. Importantly, LC itself does not require any auxiliary network, additional model, or extra user-side operation. The mapper is solely an evaluation component, used by researchers—not users—to translate hedging expressions into numerical confidence scores for benchmarking and comparison. At inference time, users interact directly with the model, and LC expressions are generated with no additional modules, unlike prior methods that depend on auxiliary networks during use. Thus, our mapper serves an offline evaluation purpose rather than being part of the LC mechanism itself.

---

> ### Author Response · Authors · 2025-11-25
> **Response on Logit Access in Commercial LLM APIs and P(True)**
>
> We appreciate the reviewer’s correction and the opportunity to clarify this point. While it is true that some commercial APIs expose partial logit information, this is not uniformly the case for modern frontier models.
>
> First, several of the most widely used closed-source models, including GPT-5, according to the official documentation, do not provide logit access at all. As these models dominate real-world deployment, the absence of logits remains a practical limitation for a large fraction of users.
>
> Second, even among APIs that do expose logits, many provide only top-k logits (often top-5). This is insufficient for a number of logit-based uncertainty methods. For example, methods such as P(True) require normalized probabilities over specific candidate tokens, which in turn requires access to both P(“True”) and P(“False”). If one of these tokens falls outside the API’s top-k shortlist—as is common when the model is uncertain or the “false” token is low ranked—then the method cannot be computed reliably.
>
> For these reasons, we argue that although partial logit access exists in some APIs, logit-based uncertainty methods cannot be applied consistently across today’s most widely deployed commercial LLMs. This motivates the need for black-box approaches such as ours, which do not rely on logits at inference time.

---

> ### Author Response · Authors · 2025-11-25
> **Clarifying the Roles of Sections 3–4 vs. Section 6**
>
> Section 6 is not designed to extend Sections 3 and 4 directly, but to serve a different purpose in the overall workflow of the paper.
>  - Sections 3 and 4 introduce the infrastructure needed to study LC at all:
>     * Section 3 provides a human-annotated benchmark of hedging expressions.
>     * Section 4 provides a mapper that converts hedges to numerical confidence for quantitative evaluation.
>
> These sections supply the tools that make systematic LC evaluation possible. Without them, the analysis in Sections 5 and 6 could not be performed.
>  - Section 6 addresses a separate research question: how to improve LC so that the model can express uncertainty naturally without elicitation. To pursue this goal, Section 6 uses semantic uncertainty as a supervision signal and constructs a fine-tuning pipeline. It does not build on Sections 3–4 because its objective is not to refine the mapper or the benchmark, but to show a practical approach for enhancing LC behavior.
>
> In short: Sections 3–4 provide the evaluation tools; Section 6 explores one concrete path toward improving LC itself.
>
> We will make this claim more clear in the camera ready version.

---

> ### Author Response · Authors · 2025-11-25
> **Clarifying the Intended Claim of Section 6**
>
> The current phrasing in Section 6 is ambiguous. The intended claim is the following:
>
> 1. **The goal of Section 6 is to improve LC itself**, that is, to enable the model to express uncertainty *naturally and without any special elicitation instruction*.
>    This corresponds to improving **LC → LC(SFT)**.
>
> 2. **LC(SFT) is not proposed as a replacement for LC+.**
>    LC+ is a prompted method that *requires an additional instruction* (“hedge when uncertain”).
>    LC(SFT), by contrast, aims to make the model capable of expressing uncertainty **without any additional prompting**.
>
> 3. **What Section 6 claims is strictly:**
>    - LC(SFT) > LC (the base model)
>    - LC(SFT) requires no extra user-side operation
>    - LC+ is not the target behavior; it is an *elicitation baseline* that relies on extra instructions
>
> 4. **Section 6 does *not* claim that LC(SFT) outperforms LC+ in all settings.**
>    Performance varies by dataset (e.g., LC+ is stronger on SimpleQA, LC(SFT) is stronger on NQ-Open).
>    This is expected, because the goals differ:
>    - **LC+** measures *prompted capability*
>    - **LC(SFT)** aims for *unprompted natural behavior*
>
> 5. **The contribution of Sections 3–4 remains essential.**
>    They provide the *evaluation tools* (dataset + mapper) that allow us to:
>    - measure LC, LC+, and LC(SFT) quantitatively
>    - compare them systematically
>    - identify where LC can be improved and verify whether LC(SFT) succeeds
>
> Without Sections 3–4, none of the comparisons or conclusions in Section 6 would be possible.

---

> ### Author Response · Authors · 2025-11-25
> **Clarifying What “Diverse” Refers to in the LC Dataset**
>
> The dataset is not intended to be diverse in terms of question variety. Instead, our use of “diverse” refers to the **linguistic diversity of uncertainty expressions**. For each question, we generate **50 different expressions covering five confidence levels**. Crucially, **even within the same confidence level**, the hedging strategies differ substantially—for example, modal verbs (“might”, “could”), stance markers (“I believe”, “I suspect”), evidential cues (“as far as I know”), softeners (“it seems that”), and many others.
>
> This means that the dataset captures broad variability in *how* uncertainty is expressed linguistically, rather than relying on a fixed phrase list. Such intra-level diversity is essential for evaluating mapping methods and studying LC behavior across different ways of hedging.
>
> While expanding question variety can certainly enhance downstream utility, the dataset’s primary contribution lies in providing **rich and varied hedging expressions within and across confidence levels**, which is independent of the specific question set.

---

> ### Author Response · Authors · 2025-11-25
> **P(True) method clarification**
>
> P(True) in its original form (Kadavath et al., 2022) is indeed a logit-based method.
>
> The statement in our paper refers to the sampling-based variant used in later work (e.g., Detecting hallucinations in LLMs using semantic entropy), where P(True) can also be approximated through multiple sampled generations when logits are unavailable.
>
> We will clarify this distinction.

---

> ### Author Response · Authors · 2025-11-25
> **Title**
>
> Thank you for pointing out the issue. The title will be corrected to “Can Large Language Models Express Uncertainty Like A Human?” to fix the grammatical issue.

---

> ### Author Response · Authors · 2025-11-25
> **Evaluation on Reasoning-Heavy Dataset**
>
> Thank you for pointing this out. To assess whether our findings extend beyond short-form QA, we additionally evaluated LC and LC (SFT) on a **reasoning-heavy subset of MMLU** (including categories such as formal logic, high-school mathematics, and computer science). We observe consistent improvements from our fine-tuning framework on this subset as well: for example, LC’s ECE decreases from 0.421 to 0.249, while AUROC increases from 0.503 to 0.744 after SFT. This suggesting that the proposed framework generalizes not only across datasets, but also to more reasoning-centric evaluations. We will include this new experiment and the detailed results table in the Appendix.

---

> ### Author Response · Authors · 2025-11-28
>
> Dear reviewer, thank you for your comprehensive feedback. We have addressed most of the concerns raised in your review. We noticed many of your points align with those of Reviewer MG3s, who has since increased their score after viewing our reply. Please let us know if there are any remaining questions or concerns you would like us to clarify.

---

### Author Response · Authors · 2025-12-01
**Review Summary — Score Change from (6,4,4 → 6,6,4) Before the Incident**

**Dear AC:**

Because ACs were reassigned after the recent OpenReview incident, we provide this brief summary of reviewer concerns and how they were addressed **before** the incident.  **Importantly, one reviewer (MG3s) raised their score from 4 → 6 prior to the incident**, after reading our clarifications and additional experiments.

---

### **1. Timeline**
- **Nov 25:** Rebuttal submitted.
- **Nov 26:** Reviewer **MG3s** raised score **4 → 6** after confirming their concerns were resolved.
- **Before the incident:** Reviewer **o6xa did not** reply, but their main concerns overlap with MG3s and were addressed.
- **Nov 28:** OpenReview incident announced.

---

### **2. Reviewer MG3s (score raised 4 → 6)**
- **LC vs. LC+ motivation** → clarified LC as natural setting; LC+ as diagnostic tool; mapper as offline evaluator.
- **Dataset subjectivity** → clarified linguistic diversity (50 hedges/question) and population-level filtering (>12k annotations).
- **Generalization** → added NQ-Open and MMLU reasoning results showing consistent improvement.

---

### **3. Reviewer o6xa (score = 4, no reply before incident)**
Main concerns (all addressed in rebuttal):
- **Roles of LC, LC+, mapper, Section 6** → clarified workflow and purpose.
- **Dataset “diversity”** → clarified that diversity refers to hedging variety, not question variety.
- **Generalization** → addressed with MMLU reasoning results.
- **P(True) / baseline details** → clarified logit-access limits and corrected descriptions.

---

### **4. Reviewer wG75 (score = 6)**
- Concerns on **annotation filtering**, **stability**, **rater instructions** → resolved via population-level filtering, adding bootstrap CIs, and clarifying rater workflow.
- Minor issues on **decisiveness scores** and **title wording** → corrected.

---

### Meta-Review · Area_Chair_AEr4 · 2025-12-03

**Summary:**

This paper releases a dataset containing hedging expressions along with a method for producing confidence scores from hedging expressions.  Equipped with these two, the authors conduct a study, showing that existing LLMs can be prompted for calibration in this setting.  The authors further propose a fine-tuning algorithm to improve linguistic confidence.  Reviewers point out the following concerns: (1) paper contains untrue criticisms of competing methods so that LC+ is actually not so novel or useful compared to baselines, (2) unclear contributions of sections 3 and 4 which may at least in part stem from poor presentation, (3) dataset only contains questions from SimpleQA, (4) potentially weakened baselines, (5) lack of error bars, (6) few annotators per sample

**Reviewer Concerns:**

(3) was addressed with MMLU, and the authors also clearly addressed (6).  The authors claim that multiple reviewers misunderstood their contributions and the point of LC+, even for thorough reviewers, which suggests a lack of clarity in the draft.  In general, I think (1) and (2) are substantial points which have not been addressed sufficiently.

**Reviewer Scores:**

The scores started off as 4, 6, 4, with one reviewer raising their score from 4 to 6 during the discussion period.  However, I think the first reviewer’s points largely still stand, and I anticipate that reviewer maintaining their current score.  Since I also agree with that reviewer’s points, I tend towards rejection here.

---

### Decision · Program_Chairs · 2026-01-26

Reject